# DelGrad: exact event-based gradients for training delays and weights on spiking neuromorphic hardware

Julian Göltz [1,2,4] ✉, Jimmy Weber [3,4] ✉, Laura Kriener [2,3,4] ✉, Sebastian Billaudelle [1,3], Peter Lake[1], Johannes Schemmel [1], Melika Payvand [3,5] ✉ & Mihai A. Petrovici [2,5] ✉

Spiking neural networks (SNNs) inherently rely on the timing of signals for representing and processing information. Augmenting SNNs with trainable transmission delays, alongside synaptic weights, has recently shown to increase their accuracy and parameter efficiency. However, existing training methods to optimize such networks rely on discrete time, approximate gradients, and full access to internal variables such as membrane potentials. This limits their precision, efficiency, and suitability for neuromorphic hardware due to increased memory and I/O-bandwidth demands. Here, we propose DelGrad, an analytical, event-based training method to compute exact loss gradients for both weights and delays. Grounded purely in spike timing, Del-Grad eliminates the need to track any other variables to optimize SNNs. We showcase this key advantage by implementing DelGrad on the BrainScaleS-2 mixed-signal neuromorphic platform. For the first time, we experimentally demonstrate the parameter efficiency, accuracy benefits, and stabilizing effect of adding delays to SNNs on noisy hardware. DelGrad thus provides a new way for training SNNs with delays on neuromorphic substrates, with substantial improvements over previous results.

The mammalian brain has always represented the ultimate example of computational prowess, and therefore remains an important source of inspiration for understanding intelligence and replicating it in artificial substrates. In particular, its specific mechanisms for transmitting and processing information have been the subject of intense scrutiny and debate. Among these, the pulsed communication between neurons, predominantly based on all-or-none events, called action potentials or spikes, stands out as a distinguishing feature, and has thus been suggested to play an important role in the brain's remarkable combination of computational performance and energy efficiency[1,2]. Consequently, spike-based communication represents a de facto standard across current neuromorphic platforms, which aim to inherit the proficiency

of their biological archetype by replicating chosen aspects of its structure and dynamics[3–7].

Among the various encoding schemes proposed for spiking neurons, the representation of information within the specific timing of individual spikes is of particular interest[8], as it effectively allows the communication of, ideally, real-valued signals on an energy budget equivalent to only generating and transmitting a single bit. This gives rise to a specific call for SNN training algorithms that exploit the temporal richness of spike timing codes for solving computational tasks efficiently and accurately, while remaining capable of operating under the realistic constraints of the underlying physical substrate, whether biological or artificial.

[1]Kirchhoff-Institute for Physics, Heidelberg University, Heidelberg, Germany. [2]Department of Physiology, University of Bern, Bern, Switzerland. [3]Institute of Neuroinformatics, University of Zurich and ETH Zurich, Zürich, Switzerland. [4]These authors contributed equally: Julian Göltz, Jimmy Weber, Laura Kriener. [5]These authors jointly supervised this work: Melika Payvand, Mihai A. Petrovici. ✉e-mail: julian.goeltz@kip.uni-heidelberg.de; jimmy.weber@ini.uzh.ch; laurak@ini.uzh.ch; melika@ini.uzh.ch; mihai.petrovici@unibe.ch

Recent years have seen an exciting trend in this direction, showing how the performance of SNNs can be improved by optimizing various temporal parameters. Such parameters include neuronal integration time constants[9–15], adaptation time constants[16], and delay variables[17–19].

In particular, spike transmission delays have been predicted to significantly enrich the information processing capabilities of spiking networks[20,21], but specific applications to computationally demanding tasks had remained an open issue. However, recent evidence suggests that a co-optimization of synaptic weights and delays is possible and can significantly reduce the number of training parameters in an SNN, without loss of accuracy[22,23]. This finding is especially important for neuromorphic architectures that target limited on-chip memory.

Nevertheless, from an algorithmic perspective, optimizing delays in SNNs remains an ongoing research problem. Previous literature has largely focused on either exploiting heterogeneity in delay parameters, while limiting gradient-based training to the weights, resulting in selecting the suitable delays[18,23–25], or using evolutionary, not gradient-based algorithms to find delay parameters[26]. Recently, several approaches based on surrogate gradients[27] have been proposed, using convolutional kernels[22] or finite difference methods[19,28]. The underlying surrogate-gradient approach partially addresses, at the expense of both exactness and the conservative property the gradient field[29], the discontinuities of (dis)appearing spikes by smoothing out the spiking threshold. These types of algorithms operate in discrete time, which requires the storage of neuronal activities as binary vectors over the entire history of the SNN.

In addition, from a hardware perspective, there is a growing number of neuromorphic platforms that support the emulation of delays. These implementations require additional memory elements and parameter sets to retain the information of the incoming spike for a controllable amount of time. Previous implementations of on-chip delays using Complementary Metal-Oxide-Semiconductor (CMOS) technology have used digital circuits[23,30–33], active analog circuits[34–37], or mixed-signal solutions[38]. Furthermore, emerging memory technologies such as Resistive Random Access Memory (RRAM) have also been used to realize delay elements, taking advantage of their nonvolatile, small three-dimensional footprint, and zero-static-power properties[18,32]. This increasing abundance of neuromorphic substrates offering configurable delays reveals an implicit call for algorithms capable of exploiting these novel capabilities.

In this work, we present DelGrad, which, to the best of our knowledge, is the first exact, analytical solution for gradient-based, hardware-compatible co-learning of delays and weights, using exclusively spike times for the computation of parameter updates. Its hardware compatibility stems from an algorithm-hardware co-design approach: the algorithm was developed with physical hardware systems in mind. It considers the real-valued nature of spike times on analog systems, as well as system-level constraints such as low I/O bandwidth and limited on-chip memory, resulting in a method that is inherently hardware-friendly. Compared to previous approaches, this spike-time-based method simultaneously increases precision and computational efficiency, while also minimizing the required memory footprint of the model. Under DelGrad, we quantitatively study the effect of different types of delays in relation to the size and performance of SNNs. Finally, to experimentally demonstrate the efficacy of our approach, we utilize DelGrad to perform chip-in-the-loop training of a delay-based SNN on a mixed-signal neuromorphic chip.

## Results

### Training delays in SNNs with DelGrad

While spiking neural networks share their overall structure with the more well-known artificial neural networks, the intrinsic dynamics in the networks are different. In SNNs, each unit in the network is a model of a spiking neuron, i.e., communicating with binary all-or-nothing events called spikes, that carry information in their precise timing (Fig. 1). Here, we employ the leaky integrate-and-fire (LIF) neuron model, which despite its relative simplicity captures the most important properties of biological neurons and therefore often serves as a "standard model" in computational neuroscience and neuromorphic engineering[39,40]. Among these properties are foremost the spiking communication and the leaky-integrator dynamics: all input events are integrated on the membrane potential, which, after an excitation, slowly decays back to its resting state. The precise dynamics of a network of LIF neurons are determined by parameters of the neurons, but also by the inter-neuron connectivity and its parametrization, which includes synaptic weights and transmission delays.

To train the parameters of an ANN, the error backpropagation algorithm[41,42], which optimizes parameters via gradient descent, has become the de facto standard. In contrast, only recently has it become clear that in the case of SNNs the non-differentiability of spikes is, in fact, not an impediment for performing gradient-based optimizations. The developed optimization methods for training SNNs can be roughly split into two groups: approximate, surrogate gradient approaches[19,27,43,44], and methods that employ exact spike time gradients[24,45–48]. In the following, we base our study on the exact gradient methods described in ref. 45.

The subthreshold dynamics of the membrane potential $u_m$ of an LIF neuron with exponential current-based synapses are governed by the differential equation

$$\tau_m \dot{u}_m(t) = [E_\ell - u_m(t)] + I_s(t)/g_\ell \tag{1}$$

with neuronal time constant $\tau_m$, leak potential $E_\ell$, leak conductance $g_\ell$ and synaptic input current $I_s$ defined as

$$I_s(t) = \sum_i \Theta(t - t_i) w_i \exp(-(t - t_i)/\tau_s) \tag{2}$$

where $\Theta(t-t_i)$ is the Heaviside step function, $w_i$ is the weight associated with the synapse receiving a spike at time $t_i$, and $\tau_s$ is the synaptic time constant. $I_s$ thus outputs a current which is a first-order low pass filter of the input spike train, represented by the exponential kernel $\exp(-(t - t_i)/\tau_s)$. $I_s$ is itself further leaky integrated by the neuron's membrane, with time constant $\tau_m$. Upon crossing the spiking threshold $\vartheta$, the membrane is reset to $V_{reset}$ for a refractory period $\tau_{ref}$, during which the neuron does not react to any further input spikes, and the neuron emits an output spike.

The response function of a neuron, thus, fundamentally, maps a sequence of input spike times $t_i$ to a sequence of output spike times $T_i$. Here, we focus on a single output spike per neuron for ease-of-notation, but the method can be straightforwardly extended to multi-spike scenarios, as described in Section SI.D. For one such output spike time $T$, under a parametrization given by the synaptic weights $w_i$ of the incoming connections, we can write $T$ as a function of the set of input weights $\{w_i\}$ and set of input spikes $\{t_i\}$:

$$T(\{t_i\} \cup \{w_i\}) . \tag{3}$$

Under certain conditions, depending on the values of the neuronal and synaptic time constants $\tau_m$ and $\tau_s$, the function $T$ becomes analytic, as discussed in ref. 45.

For example, for $\tau_m = \tau_s$

$$T = \tau_s \left\{ \frac{b}{a_1} - \mathcal{W} \left[ -\frac{g_\ell \vartheta}{a_1} \exp\left(\frac{b}{a_1}\right) \right] \right\} \tag{4}$$

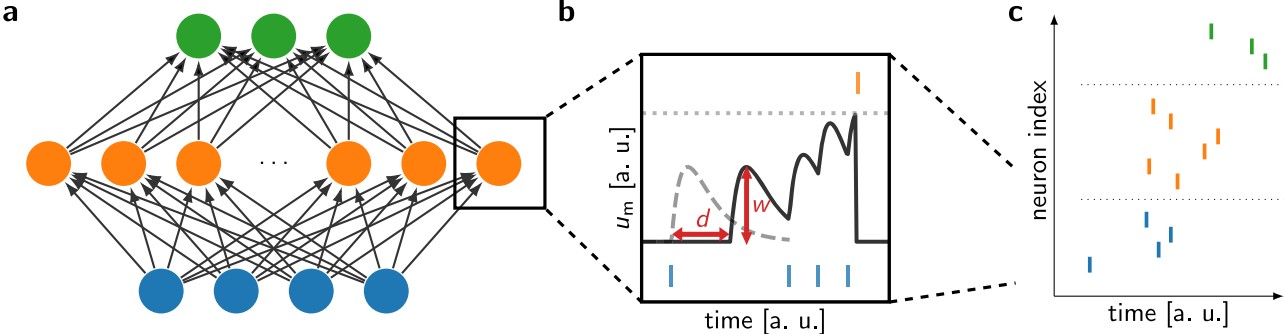

**Fig. 1 | Information flow in a (SNN). a** Network architecture of a feed-forward SNN with a spiking input layer at the bottom, a hidden layer in the middle and the output layer on top. While the methods described in this manuscript are applicable to many different network architectures, the structure depicted in (**a**), with a variable size of the hidden layer, is used in the following. **b** Zoom in on the information processing in a single leaky integrate-and-fire (LIF) neuron in the hidden layer. Incoming spikes (blue, bottom) are integrated by the neuron's membrane $u_m$ and generate postsynaptic potentials (PSPs), which accumulate additively. Once the membrane potential passes a threshold (gray dashed line), an output spike (orange, top) is generated and passed on to the neurons in the next layer. The PSP

amplitudes are modulated by the respective synaptic weights $w$ (vertical red arrow); these are the parameters that are conventionally adapted during learning. Learnable transmission delays $d$ (horizontal red arrow) shift PSPs in time, providing additional temporal processing power to the neuron. **c** Zoom out to a raster plot of the full spiking activity in the network. The information passed between the layers is encoded in the timing of the spikes. As sketched in the raster plot, in the experiments in this manuscript, we employ TTFS coding, i.e., each neuron spikes only once, however our method also generalizes to multi-spike scenarios (Section SI.D) if required by the task.

and for $\tau_m = 2\tau_s$

$$T = 2\tau_s \ln\left[\frac{2a_1}{a_2 + \sqrt{a_2^2 - 4a_1 g_\ell \vartheta}}\right] \quad (5)$$

where $a_i$ and $b$ are explicit functions of $w_i$, $t_i$ and $\mathcal{W}$ is the Lambert W function (see Eq. SI.5). Writing the spike time in this way enables us both to perform an efficient, event-based forward pass and to train this network, by calculating the exact gradient of the output of the network with respect to the network parameters.

In a multi-spike scenario (Section SI.D), all subsequent spikes after the first spike can be calculated by taking the reset into account and solving the equation for different initial conditions. Ultimately, this results in similar expressions as Eqs. (3–5).

To optimize the network parameters via gradient descent, we base the update of each parameter $\theta$ on its influence on the loss $\mathcal{L}$, the gradient $\partial\mathcal{L}/\partial\theta$. Employing the chain rule, this gradient is iteratively composed of $\partial T/\partial\theta$ and $\partial T/\partial t_i$, i.e., derivatives of the above equations.

Specifically for a network with parameters $w_i$, $\partial T/\partial w_i$ allows us to link a deviation in an output spike time to a change in weight parameters, while $\partial T/\partial t_i$ relates this deviation in the output to a deviation in the input, thereby enabling us to propagate an error in the spike time backwards through the neuron. Crucially, just like the original Eqs. (5 and 4), and in contrast to surrogate-gradient-based approaches, these derivatives only depend on spike times and parameters of the network and can be computed without the calculation or measurement of the membrane potential. This allows us to perform a fully event-based forward and backward pass, without any need for temporal discretization of forward or backwards dynamics.

Transmission delays of spike signals can now simply be introduced as additive parameters $d$ to the original spike times $t_i^\emptyset$:

$$t_i^d = t_i^\emptyset + d_i \quad (6)$$

These delayed spike times then become the relevant input for the postsynaptic neuron. As above, derivatives of this expression provide the necessary quantities for adapting the delays and for back-propagating the spike timing errors. In this case, the corresponding

equations are trivial:

$$\frac{\partial t_i^d}{\partial t_i^\emptyset} = 1 \quad \text{and} \quad \frac{\partial t_i^d}{\partial d_i} = 1. \quad (7)$$

Treating spike times as continuous variables, different from the time-binning performed in other approaches[19,22], allows this natural implementation of full-precision delays as well as the exact and simple training of the delay parameters. We note that these considerations do not depend on a specific network setup and thus apply to any activity pattern in arbitrary spiking networks. In the following, we focus our attention on the particular problem of pattern classification, for which we employ a specific network architecture and spike coding scheme (Fig. 1).

To take advantage of a well-established architectural paradigm, we now consider information propagation in hierarchical feed-forward networks. As also shown in the corresponding computational graph (Fig. 2a, solid black arrow), the input $\boldsymbol{t}^0$ is passed through the sequence of layers until it reaches the output (we use bold symbols to denote non-scalar variables). The gradient of the chosen loss function $\mathcal{L}$ then goes backwards through the network (dashed red arrow) for optimizing the parameters. In the forward pass, the only information that is transmitted is spike times $\boldsymbol{t}^l$; in the backward pass, we transmit the gradient of the loss function $\partial\mathcal{L}/\partial\boldsymbol{t}^l$, but note that it is also only evaluated at the times when neurons spike.

For SNNs with delays, the computational graph differentiates between two types of layer: neuron layers and delay layers (Fig. 2). Both layers receive input spikes $\boldsymbol{t}^{l-1}$ and return output spikes $\boldsymbol{t}^l$, but using different forward transfer functions, as given by Eq. (4)/Eqs. (5 and 6), respectively. In the backward direction, they pass the partial derivative $\partial\mathcal{L}/\partial\boldsymbol{t}^{l-1}$ discussed above.

Figure 2b, c highlight the similarity of the two layer types: both neuron and delay layers take spike trains as an input and produce spike trains as an output in the forward pass, and propagate gradients of the loss with respect to the corresponding spike times in the backward pass. Their respective computations are carried out sequentially, as depicted in Fig. 2a, with delay layers stacked in between neuron layers.

In Fig. 3a we distinguish between different types of delays: axonal delays $d_{axo}$ on a neuron's output, dendritic delays $d_{den}$ on a neuron's input, and synaptic delays $d_{syn}$ that are specific for every connection between pairs of neurons. Their respective natural representations as

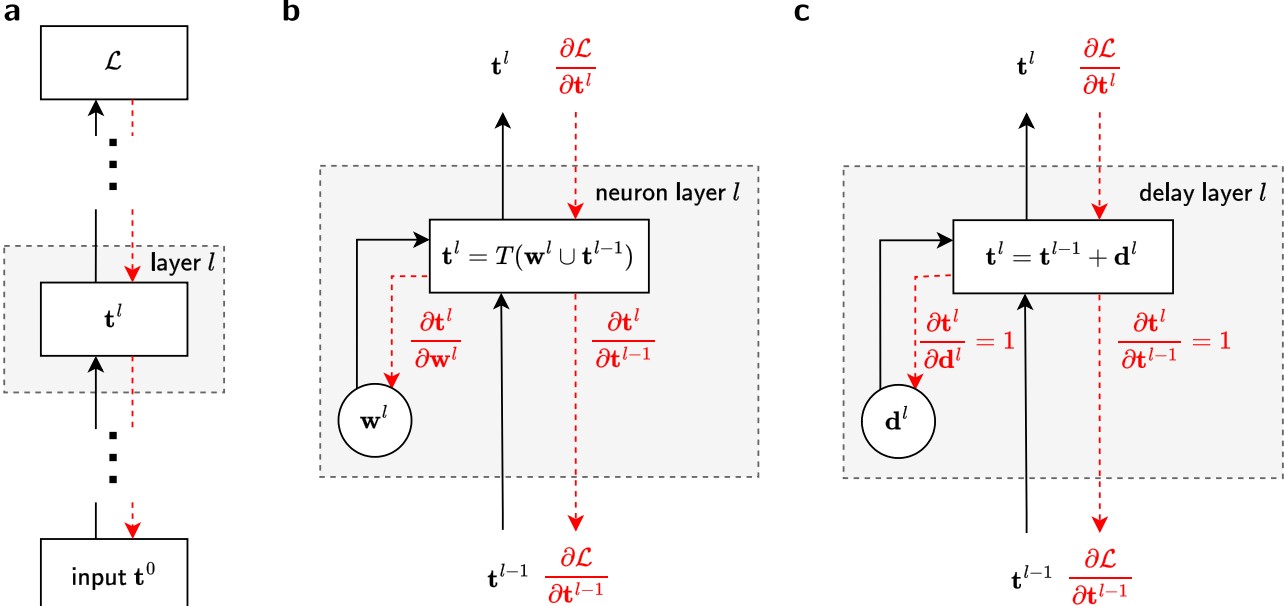

**Fig. 2 | Computational graph of a multi-layer SNN with spike-time information encoding and adjustable delay and weight parameters. a** Graph for a multi-layer network with spike times $\mathbf{t}^0$ injected into the bottom (1st) layer. In the forward pass (black arrows), each layer $l$ takes spike times as inputs and returns spike times as outputs that go into the next layer. The spike times of the topmost layer are used to compute the loss function $\mathcal{L}$. The backward pass (red dashed arrows) starts at the loss and passes the gradients backwards through the layers. We consider two types of layers: neuron layers and delay layers. **b** Neuron layer with parameters $\mathbf{w}^l$ (synaptic weights). These are used together with the input spike times $\mathbf{t}^{l-1}$ to calculate the output spike times $\mathbf{t}^l$ according to the nonlinear relation described in Eqs. (4 and 5). **c** Delay layer with parameters $\mathbf{d}^l$ that are added (linearly) to the input spike times $\mathbf{t}^{l-1}$ to calculate the output spike times $\mathbf{t}^l$ as in Eq. (6).

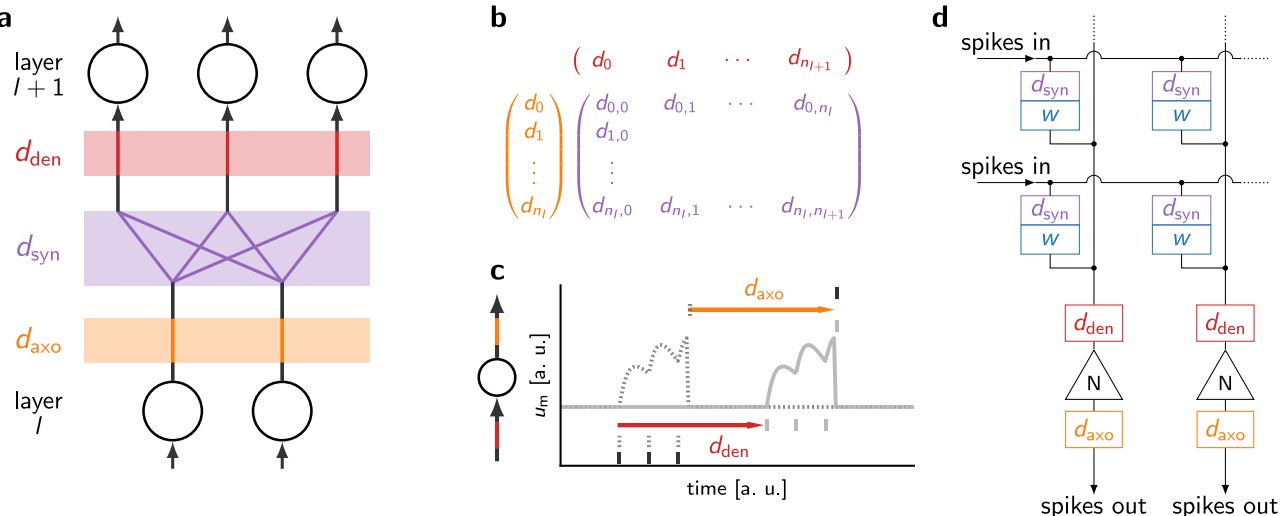

**Fig. 3 | Illustrating different types of delays. a** From bottom to top: axonal delays shift the timing of the neuron's outgoing spikes by $d_{\mathrm{axo}}$ (orange); synaptic delays shift the timing of spikes by a specific value $d_{\mathrm{syn}}$ for each pair of pre- and post-synaptic neuron (purple); dendritic delays shift the timing of the incoming spikes into a neuron by $d_{\mathrm{den}}$ (red). **b** Vector and matrix representation of the different types of delays and their dimensionality as a function of the number of pre- and post-synaptic neurons. **c** Equivalent effect of the dendritic and axonal delays on the output spike time of a neuron, due to the time-shift invariance of the temporal dynamics of a LIF neuron. **d** Schematic illustration of the location of synaptic, dendritic and axonal delay components in a generic neuromorphic crossbar architecture.

column vectors, row vectors and matrices are shown in Fig. 3b. The memory footprint of axonal and dendritic delays thus scales linearly with the number of neurons in the network, while for synaptic delays, it scales linearly with the network depth and quadratically with its width.

While in principle different types of delays can be simultaneously present in a network and can be combined with each other, it is important to note that, as illustrated in Fig. 3c, combining dendritic and axonal delays for the same neuron is redundant: as neuronal dynamics are invariant to temporal shifts, it is equivalent for the input spikes to arrive with a delay $d_{\mathrm{den}} = d$, resulting in a delayed output spike (red arrow

and gray curve), or for the output spike of the neuron to be directly delayed with $d_{\mathrm{axo}} = d$ (orange arrow and membrane dynamics in black).

Given the resource constraints of neuromorphic systems, we investigate the performance benefits incurred by the different delay types, with different requirements on the memory resources. Although a quantitative evaluation of the exact energy consumption, chip area and design complexity of different delay architectures heavily depends on system architecture and the chosen design (e.g., analog vs. digital and circuit topology), some generic statements can be made using the mathematical representation of the delay elements.

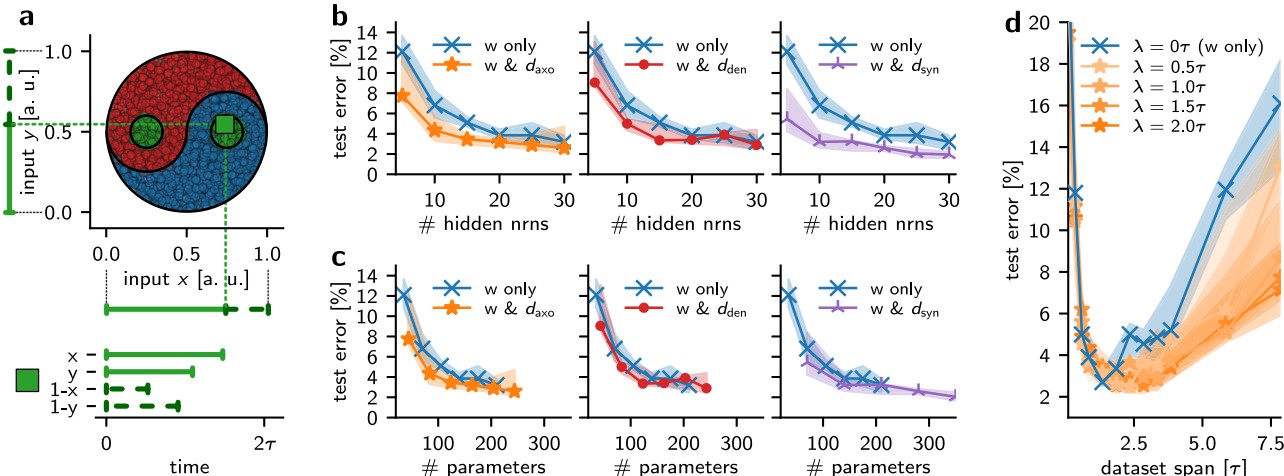

**Fig. 4 | Classification task and simulation results. a** The Yin-Yang (YY) task[50] consists of the classification of dots based on whether they belong to the Yin (red), Yang (blue), or dot (green) regions, as illustrated in (**a**). The input features are the two-dimensional coordinates $(x, y)$ of the image, along with their mirrored values $(1-x, 1-y)$, totaling four features. These features are encoded into spike times, such that a larger value of $x$ or $y$ coordinate results in a later spike time for $x$ or $y$ and an early spike time for its mirrored version $1-x$ or $1-y$ respectively. For more details on the encoding, see the original publication[50]. **b** Test error as a function of the number of hidden neurons in an SNN, using different delay types. The solid lines and markers show the median of the error, and the shaded areas illustrate the interquartile ranges (IQRs) for 10 seeds. **c** Same data as in **b** but as a function of the number of trainable parameters in the networks, i.e., counting the distinct weights and, if applicable, delays. **d** Impact of axonal delays as a function of the temporal scale of the dataset. The trainable delays cover a range $\lambda$ as indicated by the orange hue. The network performance without delays is shown in blue.

For typical crossbar architectures (Fig. 3d), the synaptic delay mechanisms are often located within the crossbar array and therefore scale with the product of the array's input and output size. In contrast, dendritic and axonal delays can be located in the periphery of the array, and thus their required area scales linearly with the input and output array size, respectively. It is worth noting that an important property of axonal delay mechanisms is that they are located directly after the neurons' output and therefore only need to operate on sparse events. In contrast, dendritic delays are located directly before the neurons' input, and after the input signals have been scaled by the synaptic weight.

Depending on the design choices, in particular on whether the synaptic integration happens in the synapses or in the neurons, this may require more complex circuitry. Note also that neurons usually receive more spikes than they emit, so the required buffering may also increase the corresponding hardware footprint of dendritic delay implementations.

## Simulation results

This section evaluates DelGrad's ability to co-train delays and weights, demonstrating improved accuracy and parameter efficiency over weight-only training. By systematically studying the effect of hidden layer size and comparing different delay types, we highlight the advantages of incorporating learnable delays.

We benchmark a PyTorch[49] implementation of the DelGrad method using the Yin-Yang (YY)[50] dataset to evaluate the impact of transmission delays on the SNN performance, and assess how this varies with the network size. This dataset is selected for its advantageous properties— compactness, making it amenable for hardware prototyping, training speed, as well as discriminatory power between network architectures and training paradigms: it leaves ample room for benchmarking above the accuracy achievable with a linear classifier. The task is to classify the region of a Yin-Yang image to which a point in the image plane belongs, as illustrated in Fig. 4a. The coordinates of the point $(x, y)$ and their mirrored values $(1-x, 1-y)$ are encoded into spike times, such that a larger value of the coordinate results in a later spike time, and an early spike time for its mirrored version.

The network architecture as shown in Fig. 1 is a feed-forward multi-layer configuration with four input neurons, followed by a variable-size neuron layer (hidden layer) and finally an output layer, comprising three neurons for the three classes (a study on deeper networks is provided in Section SI.A.1). Delay layers are inserted between neuron layers, as previously illustrated in the computational graph (Fig. 2). The neurons have no configurable biases, and the time constants are configured such that $\tau_m = 2\tau_s$. Thus, we utilize Eq. (5) for training. The refractory period $\tau_{ref}$ is set to infinity, such that all neurons only spike once. The output is represented in a time-to-first-spike (TTFS) decoding scheme, where the first output neuron to spike indicates the predicted class for a given input. To avoid negative or excessively large values for the delays, the effective delay $d$ is calculated as a logistic function of a trainable parameter $\theta_d$ such that $d = \lambda \sigma(\theta_d)$, which ensures that the delays remain bounded between 0 and $\lambda$. Further details can be found in Section SI.A.

We have chosen the time-invariant mean squared error (MSE) loss to improve accuracy and stability of training:

$$\mathcal{L}_{\Delta\text{MSE}}[\boldsymbol{t}, n^\star; \Delta_t] = \frac{1}{2} \sum_{n \neq n^\star} \left[ (t_n - t_{n^\star}) - \Delta_t \right]^2 \tag{8}$$

where $n^\star$ and $n$ denote the respective indices of the correct and wrong label neurons and $\Delta_t$ is a freely chosen parameter. The purpose of introducing $\Delta_t$ into the loss is to achieve a specific separation of $\Delta_t$ between the spike times of the correct and incorrect label neurons, instead of providing precise target spike times. To ensure a balance between model accuracy and hardware compatibility, $\Delta_t$ is set to $0.2\tau_s$ in our simulations.

We investigate the effects of different types of delay layers on accuracy, compared to configurations without any delays. Fig. 4 reports the performance of our approach on the YY dataset across different network sizes. Fig. 4b shows the percentage of misclassified samples in the test set (test error) of the network as a function of the number of hidden layers. It demonstrates that co-training delays alongside the weights always improves performance, regardless of the specific type of delay. Among the delay-augmented configurations, the variant with synaptic delays outperforms the ones with axonal- or dendritic-only parameters. This is in line with expectations, as synaptic

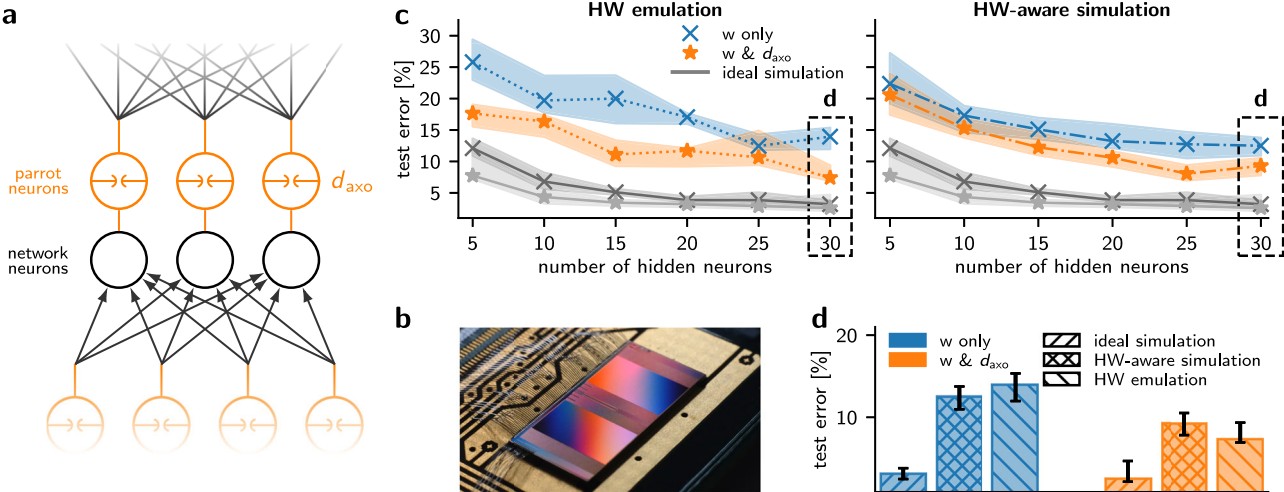

**Fig. 5 | In-the-loop training with on-chip axonal delays on BrainScaleS-2.**
**a** Schematic illustration of the network architecture for on-chip axonal delays; here, we apply this generic approach to the BrainScaleS-2 neuromorphic hardware. Each neuron in the network (black) is paired with a parrot neuron (orange) connected in a one-to-one scheme. The parrot neuron repeats each of its input spikes with a configurable delay. **b** Photograph of the BrainScaleS-2 neuromorphic chip (taken from[65]). **c** Median test errors and IQR on the Yin-Yang dataset when training network weights and axonal delays (orange) or only weights (blue). The dash-dotted lines indicate a hardware-aware simulation (cf. Section SI.A.3) and the dotted lines the hardware emulation results. For comparison, we also show the ideal software simulation results from Fig. 4b in gray. The shaded areas indicate the IQR over 10 runs with different seeds. The values for networks with 30 hidden neurons (highlighted by the dashed box) are shown for a better comparison in (**d**). **d** Detailed comparison of performances at 30 hidden neurons of an ideal simulation, hardware-aware simulation and emulation on neuromorphic hardware.

delays offer the greatest configurable parameter space among the three delay variants.

Figure 4c displays the same test errors, but now as a function of the number of parameters. This representation reveals that, at least for the YY dataset, delay-augmented networks with the same number of parameters perform similarly well, regardless of the type of delay. As before, for a given number of parameters, the co-training of delays always yields at least as good results as the training of synaptic weights alone. In other words, for the same memory footprint, a mix of both weights and delays is better than just synaptic weights.

Notably, the functional benefit of trainable delays depends to a great extent on the temporal structure of the data. In particular, we expect the training of delays to have a larger impact if the input data spans longer time scales. For YY, it is straightforward to change the temporal volume occupied by the dataset by modifying its span—the time difference between the earliest and latest possible input spikes. Fig. 4d shows the effect of trainable delays across these different spans. For small spans, errors are high because the temporal dynamics in the data are too fast for the intrinsic dynamics of the LIF neurons. However, beyond a certain point, we always observe a clear benefit of co-training delays and weights as opposed to weights alone. Furthermore, for a larger dataset span where input spikes can consequently be further apart, the range $\lambda$ of available delays that are able to push the PSPs together becomes increasingly relevant.

Optimal learning rates are determined through hyperparameter optimization for each configuration of neuron and delay layers. Across all investigated settings, our approach demonstrates robust training convergence (Fig. SI.2a) as well as exploitation of all available resources (Fig. SI.2b). Overall, these results clearly evince the added value of learning delays, as well as the ability of our algorithm to capitalize on this potential.

## Hardware results
To calculate the gradients for training weights and delays in SNNs, DelGrad only requires spike time recordings, compared to surrogate-gradient-based approaches, which also require recording membrane potential. Therefore, DelGrad is ideally suited for implementation on a variety of neuromorphic substrates, whose output is spike-based, by

design[51]. Here, we demonstrate the flexibility of our method by describing a successful application in silico, on the neuromorphic platform BrainScaleS-2 (BSS-2), that does not natively support delays.

The BSS-2 system (Fig. 5b,[7,52]) is built around a mixed-signal neuromorphic chip with 512 physical neuron circuits. The neuron dynamics are accelerated compared to biological time scales by a factor of $10^3$. The neuron circuits emulate the dynamics of the adaptive exponential leaky integrate-and-fire (AdEx) model[53] with individually configurable parameters for each neuron[54]. Neighboring neuron circuits can be connected to form multi-compartment neurons[55]. The connectivity between the neurons on the chip can be configured arbitrarily within the constraints of the two $256 \times 256$ synaptic crossbar arrays. The synaptic weights are configured digitally with 6 bit resolution.

Although the current generation of BSS-2 does not natively support on-chip delays, we present an approach that allows us to explore the computational potential of delays for the current substrate. We realize on-chip axonal delays by re-purposing a subset of the available neurons as delay elements. For this, we utilize the adaptation circuitry as well as multi-compartment functionality of the neurons on chip. This allows us to perform in-the-loop training of both synaptic weights and the on-chip axonal delays and illustrate the computational advantage obtained by the inclusion of delays. The details of the delay implementation are provided in Section SI.B.1. We additionally provide a proof-of-concept of a different on-chip realization of axonal delays using LIF neuron dynamics, which is the most widely adopted neuron model for hardware platforms (see Section SI.B.2).

Even without an explicit hardware implementation of delays, an effective axonal delay can be achieved by exploiting the dynamics of the on-chip infrastructure. For that, a "parrot neuron" is connected to the output of a neuron that is part of the actual trained network (Fig. 5a). For any spike that the network neuron produces, the parrot neuron is configured to elicit a spike after the desired delay.

In our implementation on BSS-2, this behavior is achieved via the interplay between the two neuron compartments that form a parrot neuron. The first compartment reacts to each incoming spike with a reset, which clamps the membrane voltage to the reset potential for a refractory period, during which the neuron is not responsive to incoming spikes. After the end of the configurable refractory period,

the second compartment becomes active and its adaptation mechanism almost instantly triggers an output spike of the parrot neuron. Therefore, a spike is generated after the configurable refractory period, used here as the delay. For a more detailed description of the mechanism see Section SI.B.1. We use this method, as it allows us to control the delay produced by the parrot neuron via its refractory period, which is digitally controlled on BSS-2 using 8 bits of precision. This results in a more precise and easily configurable delay, compared to using an analog variable, and is likely closer to a future implementation of native delays on a BSS-2-like system.

This delay mechanism allows us to train a network with axonal delays on BSS-2. We use an in-the-loop training approach, which means that we present a batch of inputs to the network on chip and record the spike times. The spike times are sent back to the host computer, where the loss and the backward pass are calculated in software. The resulting updates for weights and delays are then used to reconfigure the chip before the next batch is presented.

With this chip-in-the-loop setup, we train and evaluate networks, with synaptic weights alone, as well as networks that incorporate both adjustable weights and axonal delays (Fig. 5c). Similar to the simulation results presented in Fig. 4, we experimentally confirm an accuracy gain over a range of network sizes, for the networks with additional delay parameters compared to the ones with only weight parameters.

Overall, the final test errors reached in the software simulation are lower than the ones measured on hardware. This is expected, as hardware effects such as trial-to-trial variations, fixed-pattern noise and jitter on the on-chip delays disturb the dynamics. To illustrate and characterize these effects, we measured the magnitude of several noise sources found on the hardware and modeled them in a series of hardware-aware simulations. Fig. 5c shows that, when the various sources of noise are realistically modeled based on hardware measurements, the hardware-aware simulations capture the increase in test error similarly to the actual emulation. This confirms that the gap in accuracy between software simulations and hardware experiments is mostly due to the modeled sources of noise. For an in-depth description of the noise models employed in the hardware-aware simulations and an analysis of the impact of the different noise sources on the network performance, we refer to Section SI.A.3.

For an easier comparison, we focus on the most expressive networks with 30 hidden neurons, highlighted by boxes in Fig. 5c, and collect the achieved test errors in Fig. 5d, which amount to 7.40% with axonal delays and 13.95% in the weight-only case on the hardware. A full report of the achieved test errors and IQR, both in hardware-aware simulation and on chip, can be found in Table SI.1. Additionally, we note that the performance gap between the delay and no-delay setup is significantly wider on hardware than in the ideal software simulations. We hypothesize that this effect arises because the YY classification problem, by design, does not require a large network to solve, leading to few learnable parameters, low redundancy, and consequently a greater sensitivity to noise. Introducing axonal delays increases redundancy, due to the higher parameter count. However, this increase is rather small compared to the number of parameters in a weight-only setup. This suggests that the computational properties of the delays are at least partially responsible for making the network more noise resilient, explaining the larger performance gap between the two networks on noisy hardware.

These results illustrate that our method for training delays is not only applicable in ideal software simulations but can also be applied to mixed-signal neuromorphic systems. Additionally, they demonstrate the benefit of learnable delays for neuromorphic platforms, especially in resource-constrained scenarios, and might encourage the inclusion of delay mechanisms in future generations of neuromorphic systems.

## Discussion

We have introduced an exact event-based algorithm for training temporal variables, specifically transmission delays, in conjunction with synaptic weights in SNNs. Additionally, we have experimentally validated its effectiveness through both software simulations and neuromorphic hardware implementations.

Delay parameters were previously demonstrated to increase the representational power of the SNNs, even without optimization, just by training the weight parameters to select the useful delays for spatiotemporal feature detection[18,23,56]. However, this optimization-through-selection approach requires an over-allocation of resources in order to provide a sufficiently diverse set of delay parameters from which the best can be selected. To illustrate this, we compare a network of random fixed delays to a network with trained delays using DelGrad. We show in Fig. SI.3 that for the same number of delay parameters, the network with trained delays and weights has a clear accuracy advantage over the network with randomly initialized delays with weight-only optimization. Therefore, it is advantageous to combine a dedicated learning algorithm for transmission delays with hardware capable of configuring them accordingly.

Algorithms based on surrogate gradients for direct training of delay elements have been explored recently, using temporal convolution kernels[22] or numerical solutions that estimate the delay gradients using finite-difference approximations[19]. However, as pointed out in ref. 22, delay training based on finite-difference approximation[19] appears to not be sufficiently accurate to achieve an improvement over fixed, random delays. Additionally, both approaches use a time-stepped framework for calculating the gradients.

As such, delays are represented implicitly in the number of simulation time steps before transmitting a spike. However, as delay parameters are essentially shifts in individual spike times, we argue that it is more natural to have a framework where the information is explicitly represented by these spike times[32,45,46,48,57], and delays are learned as additive parameters for these times. Furthermore, the objective of building efficient asynchronous neuromorphic systems, where time represents itself, is an additional motivation for representing temporal information in spike times[3]. Such representations are naturally available from event-based sensors, where the change in the signal is encoded into spike times using the delta modulation encoding scheme[58–60].

This work brings together all the aforementioned objectives: DelGrad presents an event-based framework for gradient-based co-training of delay parameters and weights, without any approximations, and which meets the typical demands and constraints of neuromorphic hardware, as demonstrated experimentally on an analog mixed-signal neuromorphic system. As such, it takes an important step towards fully exploiting the temporal nature of SNNs for memory- and power-efficient end-to-end event-based neuromorphic systems.

In this work, we have also compared the effect of dendritic, axonal and synaptic delays on the performance of SNNs on a representative task. The synaptic delays have the highest impact on increasing the expressivity of SNNs, compared to using only dendritic or axonal delays. However, from a hardware perspective, the addition of synaptic delays imposes a quadratic growth on the size and thus the on-chip area of the network, compared to a linear growth in the case of axonal and dendritic delays. In fact, we find that when comparing the performance for equal parameter counts, the gap between different types of delays vanishes while the superiority over weight-only training persists. As memory represents one of the most critical constraints on hardware, reducing on-chip memory is of utmost importance. In particular, this means that a redesign of a chip with fewer neurons but with an intrinsic delay mechanism, based on our findings, will save energy while maintaining expressivity and performance. Therefore, our work suggests that it might be practical to consider using dendritic or axonal delays in future hardware designs, combining favorable scaling and improved processing power.

DelGrad provides an advantage in terms of hardware mappability as it only requires recording the spike times from on-chip neurons.

This is in contrast to other approaches[19,22], which need access to the membrane potential of all neurons for surrogate gradient learning[61,62]. Such voltage-based plasticity requires additional components for voltage readout, communication and potentially analog-to-digital conversion. Furthermore, this information is much denser in time than the spikes themselves, imposing further stress on the overall communication bandwidth. In both chip-in-the-loop and on-chip training scenarios, these additional requirements ultimately translate to additional circuitry; not only does this increase the complexity of the chip design, but it also inherently reduces the maximum implementable network size for a chip of a given area. Moreover, if learning is to be implemented on-chip, this additional circuitry will negatively affect the device's energy efficiency during training.

In this work, the YY dataset was used as a proof of concept and first step to benchmark our approach. The YY dataset provides a problem that can not be solved linearly and where the information can be presented using a TTFS encoding, similar to the previous work[45]. As indicated by our experiments in Fig. 4d, the performance boost provided by the inclusion of learnable delays increases when the temporal features of the data span larger time scales.

Therefore, the natural next step will reside in a more thorough benchmarking on larger datasets, and in particular on data with explicit temporal components, such as[63,64]. Especially for data provided by event-based sensors, longer time scales are required, and TTFS might reach its limits as a feasible coding scheme. Although our current software implementation only takes into account a single spike per neuron during the training, this is not a limitation of our proposed mathematical framework and training scheme (see Section SI.D). Additionally, the extension to more complex spike timing codes can go hand in hand with a shift from a feed-forward to a recurrent network architecture.

## Data availability
We used the Yin-Yang data set[50], the code is available at https://github.com/lkriener/yin_yang_data_set.

## Code availability
Code for the simulations is available at https://github.com/JulianGoeltz/fastAndDeep.

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

## Acknowledgements

We want to thank the EIS-Lab, NeuroTMA, CompNeuro, and ElectronicVision(s) groups, in particular Yannik Stradmann, Robin Heinemann, Joscha Ilmberger and Eric Müller, for the continuing support. Additionally, we are grateful to the NNPC conference 2023, where this fruitful collaboration was initialized, as well as the CapoCaccia workshop, where this work was first presented and received much helpful feedback, especially from Paolo Gibertini and Maryada. We also thank Guillaume Bellec for critical comments on the preprint and Florent Draye for providing feedback on the mathematical proofs. The presented work has received funding from the Manfred Stärk Foundation (JG, MAP), the EC Horizon 2020 Framework Programme under grant agreement 945539 (HBP; JG, LK, SB, PL, JS, MAP) and Horizon Europe grant agreement 101147319 (EBRAINS 2.0; MAP), the Deutsche Forschungsgemeinschaft (DFG, German Research Foundation) under Germany's Excellence Strategy EXC 2181/1-390900948 (Heidelberg STRUCTURES Excellence Cluster; JG, MAP), Swiss National Science Foundation Starting Grant

Project UNITE (TMSGI2-211461; JW, LK, SB, MP), and the Volkswagen-Stiftung under grant number 9C840 (SB).

## Author contributions

JG, JW, and LK jointly developed the theory, designed the experiments, implemented the code, executed simulation and hardware experiments; SB, PL, and JS contributed to the hardware experiments; MP and MAP supervised the project, contributed to the experiment design, and provided helpful guidance throughout; JG, JW, LK, MP, and MAP wrote and revised the manuscript.

## Funding

## Competing interests

The authors declare no competing interests.
