## [Transparent Peer Review file · Nature Communications]

DelGrad: Exact event-based gradients for training delays and weights on spiking neuromorphic hardware

Corresponding Author: Mr Julian Goeltz

Version 0:

Reviewer comments:

Reviewer #1

(Remarks to the Author)

Noteworthy results: This paper implements a chip-in-the-loop neuromorphic backpropagation algorithm based on the modification of synapses and temporal delays.

Significance of Results: The authors use an analytical expression for the gradient of the loss to optimize a layered neural circuit on a mixed-signal neuromorphic chip. This is a technical improvement over surrogate gradient methods that approximate the gradient. However, given the noisy results that the authors obtain when implementing the method on-chip, I still have concerns that their approach may not be significantly better than surrogate gradient methods. It does, however, only require spike timings for gradient computation, which is an advantage.

My main concern with the paper is that optimization is performed given complete spiking records off-chip (chip-in-the-loop). Most of the neuromorphic literature is dedicated to energy-efficient computation. Chip-in-the-loop methods require a standard (non-energy-efficient) computer to perform the optimization. This obviates the benefit of using the chip for energy-efficient training. However, it may still be claimed that inference using a pre-trained neural circuit will be energy-efficient. So, this slightly lessens the concern.

Support for Conclusions: The paper is very well-written and all conclusions are supported. The energy-efficiency issue is not discussed. 'Test error' was not defined in the manuscript, here I assume it means that percentage of incorrect classifications. The fact that this value is at around 20% for HW emulation seems a bit high. Test errors were not compared with other standard (non-neuromorphic) methods.

Data Analysis: The data analysis seems straightforward and I did not see any issues other than the lack of comparison with competing methods on the same dataset.

Methodology: The methodology is of high quality. The use of parrot neurons was a nice use of the neural substrate.

Sufficient Details: I believe that the methods used could be reproduced by a reader.

Comments written while studying the paper:

In my opinion, this paper should be on the hardware implementation, not DelGrad per se. I would reorganize the Abstract to refocus on the BrainScaleS-2 implementation being enabled by DelGrad. In other words, this is not a paper about computing gradients analytically, it is a paper about a large-scale hardware implementation of chip-in-the-loop machine learning that used an exact gradient computation technique.

Suggestion for new title: "Chip-in-the-loop neuromorphic machine learning on mixed-signal neuromorphic hardware with exact gradients" or something similar.

'less number of' -> 'fewer'

"recent evidence suggests that a co-optimization of synaptic weights and delays is possible and can significantly reduce the

number of training parameters in an SNN, without loss of accuracy": I don't understand this comment. Doesn't the use of delays just introduce a continuous set of parameters in time? I would believe that the number of neurons/memory used is reduced, but not the number of parameters. If not, please explain.

'operate in discrete time, which require the storage of neuronal activities' -> 'operate in discrete time, which requires the storage of neuronal activities'

"In contrast, only recently has it become clear that in the case of SNNs the non-differentiability of spikes is, in fact, not an impediment for performing gradient-based optimizations." This sentence could use the citation: Renner, Alpha, Forrest Sheldon, Anatoly Zlotnik, Louis Tao, and Andrew Sornborger. "The backpropagation algorithm implemented on spiking neuromorphic hardware." Nature Communications 15, no. 1 (2024): 9691, where surrogate gradients are used in an on-chip (not a chip-in-the-loop, but with training fully on-chip), multilayer, (Loihi 1) neuromorphic implementation of backpropagation in CMOS (not mixed-signal).

'two types of (virtual) layers' -> 'two types of (virtual) layer'

Is it the case that the neuron and delay layers are both instantiated in a single physical set of neurons? I'm not quite understanding what 'virtual' layer means, here. In other words, do delay layers add to the number of neurons used? Or is each neuron multi-use?

"Optionally, delay layers may be inserted between neuron layers, as previously illustrated in the computational graph (Fig. 2)." Why is this optional? I thought that backpropagation was implemented -via the delay layers.

'ensures that the delays remains bounded' -> 'ensures that the delays remain bounded'

'a mix a of both weights' -> 'a mix of both weights'

'parrot neurons' nice use of underlying substrate

'sources of noise sources' -> 'sources of noise'

"However, as pointed out in [22], the finite-difference approximation appears to not be sufficiently accurate to achieve an improvement over fixed, random delays." This statement seems to only be considering delay-based surrogate gradients. In the citation above, synfire-gating information coordination is able to use surrogate gradients in a different manner to get MNIST accuracies of ~97%.

Additionally, the need for chip-in-the-loop training indicates that the actual training algorithm needs a standard computer to implement. Will it be possible to transfer exact gradients to a fully on-chip algorithm? This will be important since one of the most significant uses of neuromorphic AI is energy-efficient computation. The use of a standard computer for training effectively eliminates any efficiency in the training phase. And there are many off-chip learned neuromorphic SNNs that perform very well in the inference stage.

Have the authors studied how deep of a network can be trained given their hardware noise? The BSS-2 chip may not be big enough, but this could be tested in simulation.

(Remarks on code availability)

Reviewer #2

(Remarks to the Author)

The paper focuses on training transmission delays in networks of spiking neurons with exact gradient methods, using a framework of leaky integrate-and-fire neurons and Time-to-First-Spike coding (TTFS). This is a timely topic for both theory and hardware people and the paper contributes insights for both communities. The authors show that a situation where the membrane time constant is exactly twice the synaptic time constant is of particular interest because in this case the output spike time in layer n is an analytical function of the input spike times of neurons in layer $n-1$ (Eq. 5 in the paper). This explicit formula avoids access to the derivatives of the membrane potential at threshold (and also avoids the somewhat awkward Lambert function of Eq. 4).

Overall, I like the paper. In particular I am impressed by the training results on the Brainscales hardware platform with the hardware in the loop. The idea of introducing parrot neurons to implement delays on the existing neuromorphic hardware is nice. I suggest to publish the paper after my major comment and a few minor comments have been taken into account.

Major:

The main text should stress more clearly and early on two assumptions made in the paper.

(i) I understand that the formalism can be extended to multiple spikes per neurons (as mentioned at the end of the Discussion). However, as the paper currently stands, all explanations and notations in the theory part of the main text

assume single-spike coding/Time-To-First-Spike (TTFS) coding, not only in the output layer but also in all intermediate layers. Otherwise Eqs (3-5) do not work. For example in equation (3), there should also be a dependence upon that last time of reset of the neuron in layer n ; and if the input neurons in layer $n-1$ are allowed to fire several spikes then the authors should mention more explicitly that the index i of t_i and w_i does not refer to neurons but to spikes so that a neuron with two spikes enters twice. (The assumption that refractory time is set to infinity is unfortunately only introduced in the simulation section. If the authors move this information to the theory section, the above problem will be solved).

(ii) The simulations are limited to networks with one hidden layer. This is a severe limitation. In networks with multiple hidden layers training instabilities may arise, which can be traced back to the vanishing or exploding gradient problem as suggested in Stanojevic et al. (2024). I assume that these instabilities can be controlled if gradients in the present model are evaluated AS IF the membrane potential was crossed with a slope of one. I guess, that, analogous to Eq. (5), the authors would be able to explicitly calculate the slope of the membrane potential at threshold and use this information to stabilize their algorithm for deep networks, analogous to Stanojevic et al. (The above problem could be simply solved by adding a few sentences in the discussion section on the generalization to multiple layers)

Minor comments (mostly linked to the above points)

p1 less number  a smaller number

p2/ Fig 1, Caption c: Please mention that ALL neurons in the hidden layers also fire at most a single spike, which is necessary for Eqs 3-5. Therefore not just the spike output layers is TTFS (in the sense that the first spike across all neurons counts) but each neuron in each layer has TTFS coding (in the sense that only the first spike of each neuron counts and later spikes of the same neuron are absent or ignored or suppressed by infinite refractory period).

p2 "resulting in selecting the suitable delays." An early example of the idea of delay tuning via selection is 'A neuronal learning rule for sub-millisecond temporal coding.' Nature 383 (6595), pp. 76–78 (though in an unsupervised setting rather than a supervised one).

p3 "that employ exact spike time gradients [41, 42]." May be the paper of Stanojevic which is cited later could already cited here? I leave this decision to the authors.

p3. before Eq. 3

For one such output spike time T  REPLACE BY:

Assuming that the membrane potentials are all reset at time zero, the first output spike of neuron j in the next layer ...

p5. "The refractory period τ_{ref} is set to infinity, such that all neurons only spike once." THIS assumption is used much earlier (see above) so that the sentence should be shifted to the theory section.

(Remarks on code availability)

Reviewer #3

(Remarks to the Author)

Summary:

The paper describes a method to train a feed-forward network of spiking neurons with delay. It defines a mathematical model for training on a general purpose computer, and an implementation on the mixed-signal brain scales neuromorphic hardware. The mathematical model computes analytically the exact spike-time for each neuron and adds a constant delay parameter. Backpropagation is therefore exact (as long as spiking order does not change, and no spike appears or disappears). For the hardware implementation, hardware in the loop training is used to train robustly despite imperfections of the semi-analog brain scales hardware. A simple noise model simulation is proposed to explain the reduction in accuracy after training on the hardware.

Main strength:

Developing a mathematical model that is viable for analog (or mixed hardware) is very difficult since it requires adapting to the hardware constraints. It is here well executed, and the achievement is well needed for the community since there are emerging techniques to train SNN using exact spike times but this is the first (or maybe second) demonstration on hardware which scales. Without a doubt it is the first to do that with delay parameters and prove on the hardware that delays can be used (and are useful) for a hardware implementation.

The writing is quite good, all the results are very clearly described, and I am convinced that the results are reproducible.

Main weaknesses:

1) Unfortunately the dataset is rather small and easy (probably significantly easier than MNIST for instance). It means that

the analysis (synaptic versus axonal etc...) is probably not very informative on larger problems where depth would create more problems. Keeping the dataset small is totally understandable for the hardware implementation since the hardware dimensions are adding constraints, but I wonder if the claim that delay parameters are bringing benefits would be also true on a deeper network for instance (this is a very general speculation see comments 3 and 4, for more precise comments). To be clear I am not asking to try another dataset in a potential revision, rather, acknowledging that the dataset is small but adapted to the hardware prototype is sufficient. In this logic, any general claim on the quality of the mathematical model for neural network training should have a caution that it is only shown here for a small network and dataset.

2) I find that the intro is not doing a very good job at positioning this paper in comparison with the literature. Reading the intro it is sometimes ambiguous if the authors claim that exact backpropagation in spiking is a novelty of this paper for instance. Here are recommendations:

2a) Conceptually, exact backprop for SNN was published in Bohte 2002. This ref is present but it should also be cited along with [41] and [42] in the intro. I find it interesting that in 20 years, this technique was forgotten and rediscovered. I wonder if the authors have something to add about that. Maybe it works better now because the neuron model changed? the dataset encoding? (Caution: I expect general speculation here, it is better not writing more rather than a half-convincing statement. For instance, it's not just about delays, given that many papers have shown that harder datasets are solvable without delay).

2b) The intro often refers to "the true gradient" but fails to report that there are many discontinuities in the network computation (example: after a weight update it probably occurs often that the output spike of a neuron arrives before it's input, or that a new spike is added). It would be great to acknowledge that the gradient computation, like others, is not solving this. Optionally: writing that surrogate gradient is not only inexact but also complementary since it addresses this partially, could be useful.

2c) The last paragraph of the intro: "In this work, we present DelGrad ..." would deserve some clarifications. When parsing the first sentence it is hard to know what the authors claim as the theoretical novelty. Is that the intersection of exact spiking and application to parametrized delay? or is it the demonstration that it can work on hardware (which I find a lot more impressive). This claim on memory foot-print is also not very convincing since the dataset size is meant to match the hardware size, so the CPU / GPU simulation is certainly not in a regime where memory foot-print is a real bottleneck, and therefore I would not be convinced that the authors have strong results to support that. In my opinion, it would be wiser to highlight that the study describes a dense analysis of different implementation strategies for delays in SNN and focus on the hardware part -- rather than claiming strong general software results which are weakly supported.

Main questions or suggestions:

3) There seems to be an interesting emerging property of axonal delays. I suspect that adding a constant axonal to all neuron would have only benefits. It does not affect the gradient with respect to d or lower layers, but it avoids some of the gradient discontinuities: with sufficiently large axonal delay, the output neuron spike would never arrive before its inputs, thus avoiding at least this type of discontinuities. It seems that the authors did not try to add a global fixed axonal delay (not trainable), but I am guessing that the delays are currently initialized randomly (uniformly?) in the reachable interval $[0, \lambda * \tau]$. So with large λ , this effect could still emerge when most delays are large enough. This could explain that large λ stabilizes the training for all cases in Figure 4d (it seems true for all values of τ). Therefore, I wonder if that benefit would also be happening if axonal delays are simply lower bounded, or all having the same high constant value (not necessarily trainable)? If yes I wonder if the gap in performance brought by delays could be explained by that effect. Or is there another (and better supported) intuition on why delays are helping?

Making new simulations with lower bounded delays (or constant) might be difficult for the authors, so discussing this verbally (at least in the rebuttal) might be convincing enough for me. As a related side comment, I wonder if the influence of λ would be more readable in Figure 4d if the x-axis would be λ and the color code would be τ . Currently the shades of orange are all overlapping.

4) I enjoyed the rationale that chip area savings makes axonal/ dendritic delays better than synaptic delay. I also read that delay (refractory period) is quantized on 8-bit and weight strength quantized on 6-bit. Is it because neuron parameters can take more chip area and have therefore more precision? I also wonder, the voltage threshold or neuron bias is also quantized on 8-bit? I wonder whether there is something inherent in the physics of "hardware delay computation" that could make that encoding/implementation better than a weight encoding. I have no idea about this, but I would love to hear whether some different scaling rule emerge in terms of chip-area versus accuracy trade-off between delays and synaptic weights or threshold parameter.

5) Why stop around 35 hidden neurons if the crossbar array has a 256 x 256 size ? Any reasons to explain these dimensions would be useful.

(Remarks on code availability)

The code is readable and seems consistent with the reported results.

I did not run the code.

Version 1:

Reviewer comments:

Reviewer #2

(Remarks to the Author)

Overall, the authors addressed the points I had raised. Regarding the details, I still have three minor problems.

1) For some reason, the citation to the paper of Stanojevic did NOT make it onto page 3 line 178 as promised by the authors in their response to referee.

2) In caption of Fig. S18 it should probably read horizontal and not vertical bar.

3) The unpublished github reference in the reply to referee 2 DOES NOT show that the gradient travels back to the first layer. First, MNIST is known to be supereasy as a task - multiple layers are not needed. (see, e.g., Illing et al., 2019) Second, and more importantly here, if you overparameterize each hidden layer, you are in the so-called LAZY regime which is the regime theoretically explored in the paper on 'Neural Tangent Kernels' (NTK). Essentially, the theory says that in an overparameterized network the representation at initialization is rich enough that you only need to move a few weights by a small amount to learn the desired task. One way to do this, is to move only the output weights.

You can avoid lazy networks by choosing a very weak initialization (close to zero) so that many weights have to move. Lots of literature on that from the theory community. You can start with the paper of van Meegen and Sompolinsky (2025) or start from Saxe and Summerfield and look at papers cited there.

If you want to show learning in deep networks, you have to use tasks where depth is known to be important.

lazy and non-lazy:

Jacot A, Gabriel F, Hongler C. Neural tangent kernel: Convergence and generalization in neural networks. 2018. pp. 8571–8580.

Chizat L, Oyallon E, Bach F. On Lazy Training in Differentiable Programming. NeurIPS. 2018. Available: <http://arxiv.org/abs/1812.07956>

Flesch, ... Saxe, Summerfield, Neuron doi: 10.1016/j.neuron.2022.01.005

Meegen, Sompolinsky, <https://www.nature.com/articles/s41467-025-58276-6>

Illing et al. 2019 <https://www.sciencedirect.com/science/article/pii/S0893608019301741?via=ihub>

(Remarks on code availability)

Reviewer #3

(Remarks to the Author)

I would like to thank the authors for addressing all my comments seriously.

I was already favorable to the publication of this article and I continue to think it will be useful for the community.

(Remarks on code availability)

Response to the reviewers

Dear Reviewers,

We sincerely thank you for your insightful and constructive feedback on our manuscript. Your comments have helped us to significantly strengthen the paper, both in clarity and in content.

In particular, your suggestions led to several important improvements. We have extended and clarified the theoretical framework, including a new supplementary section that formally treats the multi-spike case (see Section SI.D), while keeping the main text focused and accessible through the single-spike formulation. We have also addressed questions regarding the depth of trainable networks by performing new simulations with deeper architectures, both in idealized and hardware-aware settings (see Section SI.A.1). Additionally, we have elaborated on our motivation for using the Yin-Yang dataset and better situated our method within the state of the art (see Section SI.A.4). Finally, we followed the recommendation to better articulate the hardware focus of the paper in the title: “DelGrad: Exact event-based gradients for training delays and weights on spiking neuromorphic hardware”

Below, we reply to each concern of the reviewers one by one. We have addressed the changes in the main and supplementary texts, highlighted with a strike-through red when text was removed, and in blue when text was added. The version of our manuscript showing these changes also features line numbers, which we reference when addressing our changes in the following.

Thank you again for your thoughtful feedback and for the opportunity to improve our work.

Reviewer 1

Noteworthy results: This paper implements a chip-in-the-loop neuromorphic backpropagation algorithm based on the modification of synapses and temporal delays.

Significance of Results: The authors use an analytical expression for the gradient of the loss to optimize a layered neural circuit on a mixed-signal neuromorphic chip. This is a technical improvement over surrogate gradient methods that approximate the gradient. However, given the noisy results that the authors obtain when implementing the method on-chip, I still have concerns that their approach may not be significantly better than surrogate gradient methods. It does, however, only require spike timings for gradient computation, which is an advantage.

Reviewer Point P.1.1 — My main concern with the paper is that optimization is performed given complete spiking records off-chip (chip-in-the-loop). Most of the neuromorphic literature is dedicated to energy-efficient computation. Chip-in-the-loop methods require a standard (non-energy-efficient) computer to perform the optimization. This obviates the benefit of using the chip for energy-efficient training. However, it may still be claimed that inference using a pre-trained neural circuit will be energy-efficient. So, this slightly lessens the concern.

Reply:

You are correct that for in-the-loop training scenarios inference is the main target of claims on energy-efficiency. Although that by itself encompasses a large body of low-power sensory processing application scenarios, on-chip training brings a significant leap in terms of energy efficiency during training. And indeed, there is a large focus on on-chip training in the neuromorphic community. We think that our approach would be beneficial for training networks on-chip, too, since the general advantages over surrogate-gradient-based approaches persist: Also in an on-chip training scenario surrogate-gradients require infrastructure to record, buffer and communicate the membrane voltages of all neurons simultaneously to the processing unit

that calculates the parameter updates. This additional infrastructure will not only require energy to operate but in addition consume chip area that could be used otherwise.

To clarify these points in the manuscript we have reworked the “Hardware mapping” paragraph in the discussion (page 10, line 658):

DelGrad provides an advantage in terms of hardware mappability as it only requires recording the spike times from on-chip neurons. This is in contrast to other approaches [19, 22], which need access to the membrane potential of all neurons for surrogate gradient learning [62, 63]. Such voltage-based plasticity requires additional components for voltage readout, communication and potentially analog-to-digital conversion. Furthermore, this information is much denser in time than the spikes themselves, imposing further stress on the overall communication bandwidth. In both chip-in-the-loop and on-chip training scenarios, these additional requirements ultimately translate to additional circuitry; not only does this increase the complexity of the chip design, but it also inherently reduces the maximum implementable network size for a chip of a given area. Moreover, if learning is to be implemented on-chip, this additional circuitry will negatively affect the device’s energy efficiency during training.

Support for Conclusions: The paper is very well-written and all conclusions are supported.

Reviewer Point P.1.2 — The energy-efficiency issue is not discussed.

Reply:

We hope to have addressed this issue in our reply above (Point P.1.1).

Reviewer Point P.1.3 — ‘Test error’ was not defined in the manuscript, here I assume it means that percentage of incorrect classifications.

Reply:

Thank you very much for pointing this out. We have added a definition at the first occurrence (page 7, line 397).

Reviewer Point P.1.4 — The fact that this value is at around 20% for HW emulation seems a bit high.

Reply:

Indeed, as you have noticed in Fig. 5c, a test error of 17.65% is high. Like for other datasets, the achieved test error on Yin-Yang depends strongly on the network size. We report results across a range of sizes, over which the performance improves significantly. The larger networks, with 30 hidden neurons, achieve a test error as low as 7.4%. In contrast, the high test error values of 17.65% occur only in the smallest networks with 5 hidden neurons. This effect is due not only to the limited representation capacity of small architectures, but also to their vulnerability to hardware noise. The Yin-Yang dataset is explicitly designed to be very hard to solve under the impact of noise (because the points to be classified can be arbitrarily close to the decision boundaries, for details see [48]). With fewer neurons, the redundancy in the network is reduced which directly lessens the ability to average out the impact of trial-to-trial variability and to cover for neurons that behave suboptimally due to fixed-pattern noise.

For the analogy to the ideal simulations, we measured the performance on hardware for the same network sizes (5 to 30). But in order to direct the focus of the reader onto the networks that are expected to perform well under hardware constraints, we have now highlighted the 30-neuron network in Fig. 5c and explicitly stated the achieved errors in the main text.

Reviewer Point P.1.5 — Test errors were not compared with other standard (non-neuromorphic) methods.

Reply:

Thank you for pointing this out. We agree. We have added a figure to the SI which visually compares our results on the Yin-Yang dataset to the ones presented by other publications including the ANN baseline (see Fig. SI.5). The plot distinguishes between results achieved with delay-based and weight-only SNNs as well as simulation-based and hardware results.

Reviewer Point P.1.6 — Data Analysis: The data analysis seems straightforward and I did not see any issues other than the lack of comparison with competing methods on the same dataset.

Reply:

Thank you! For the lack of comparison, we hope our previous point resolves that.

Methodology: The methodology is of high quality. The use of parrot neurons was a nice use of the neural substrate.

Sufficient Details: I believe that the methods used could be reproduced by a reader.

Comments written while studying the paper:

Reviewer Point P.1.7 — In my opinion, this paper should be on the hardware implementation, not DelGrad per se. I would reorganize the Abstract to refocus on the BrainScaleS-2 implementation being enabled by DelGrad. In other words, this is not a paper about computing gradients analytically, it is a paper about a large-scale hardware implementation of chip-in-the-loop machine learning that used an exact gradient computation technique.

Suggestion for new title: "Chip-in-the-loop neuromorphic machine learning on mixed-signal neuromorphic hardware with exact gradients" or something similar.

Reply:

While we agree that the hardware implementation on BrainScaleS-2 is an important part of the paper, we would like to emphasize in the abstract that this paper contains three important contributions:

- the derivation of an exact, purely event-based framework that can be used to train delays and weights simultaneously in an SNN
- the considerations about the different delay types and their performance-efficiency trade-offs, especially in the context of different types of hardware implementations
- the demonstration of the hardware amenability of our method by training weights and delays on BrainScaleS-2

Nevertheless, we agree that emphasizing the hardware connection already in the title is important and we have adapted the title to "DelGrad: Exact event-based gradients for training delays and weights on spiking neuromorphic hardware".

Reviewer Point P.1.8 — 'less number of' → 'fewer'

Reply:

Fixed, thanks.

Reviewer Point P.1.9 — "recent evidence suggests that a co-optimization of synaptic weights and delays is possible and can significantly reduce the number of training parameters in an SNN, without loss of accuracy": I don't understand this comment. Doesn't the use of delays just introduce a continuous set of parameters in time? I would believe that the number of neurons/memory used is reduced, but not the number of parameters. If not, please explain.

Reply:

Indeed, we do mean parameters and not simply neurons. In Fig. 4c we consider the total number of parameters, i.e., the total number of delays and weights, each of them counting as one parameter. The figure shows that by introducing parameters of a different nature (delays as opposed to weights), it is possible to conserve performance while using a smaller total number of parameters. References [23, table 1] and [22, table 2] corroborate these findings.

Reviewer Point P.1.10 — 'operate in discrete time, which require the storage of neuronal activities' → 'operate in discrete time, which requires the storage of neuronal activities'

Reply:

Thanks, fixed.

Reviewer Point P.1.11 — "In contrast, only recently has it become clear that in the case of SNNs the non-differentiability of spikes is, in fact, not an impediment for performing gradient-based optimizations." This sentence could use the citation: Renner, Alpha, Forrest Sheldon, Anatoly Zlotnik, Louis Tao, and Andrew Sornborger. "The backpropagation algorithm implemented on spiking neuromorphic hardware." *Nature Communications* 15, no. 1 (2024): 9691, where surrogate gradients are used in an on-chip (not a chip-in-the-loop, but with training fully on-chip), multilayer, (Loihi 1) neuromorphic implementation of backpropagation in CMOS (not mixed-signal).

Reply:

Thanks for the reference, we have added it in the next sentence, where we can highlight the important distinction between surrogate-gradient and spike-time-based approaches more easily.

Reviewer Point P.1.12 — 'two types of (virtual) layers' → 'two types of (virtual) layer'

Reply:

Thanks, fixed.

Reviewer Point P.1.13 — Is it the case that the neuron and delay layers are both instantiated in a single physical set of neurons? I'm not quite understanding what 'virtual' layer means, here. In other words, do delay layers add to the number of neurons used? Or is each neuron multi-use?

Reply:

We have clarified the formulation and in particular removed the word “virtual” to avoid confusion (page 4, line 270, caption of Fig. 2). This part of the text is focused on the simulation of our model and our PyTorch framework. Within this framework (see Fig. 2) we implement the neurons as a PyTorch layer that process the incoming spike times non-linearly. The delays are, for technical reasons, implemented as another layer (separated from the neurons) that acts linearly on the spike times. With this framework we can stack the delay and neuron layers, or we can model a weight-only network by stacking neuron layers without delay layers in between. Note that when we count the number of hidden layers, we count the number of neuron layers, as those are the ones that apply a non-linear transformation to the spike times. The number of hidden neurons as, e.g., depicted on the x-axis in Fig. 4 is determined by the width of the neuron layers.

Concerning the re-using of neurons: In our experiments on BrainScaleS-2 we use neuron circuits to implement delays. This is a hack that is necessary because the chip does not have dedicated delay circuitry. We do not re-use neuron circuits that are used as actual network neurons for the delay creation, because our networks are small enough to leave enough remaining unused neuron circuits that we can employ as delays. In principle, if there weren't enough neuron circuits, it might be possible to time-multiplex the emulation and re-use neuron circuits that were part of the network also for the delay creation. Note however, that this would slow down emulation drastically, as the neurons would need to be reparameterized every time they switch roles from network neurons to delay.

Reviewer Point P.1.14 — “Optionally, delay layers may be inserted between neuron layers, as previously illustrated in the computational graph (Fig. 2).” Why is this optional? I thought that backpropagation was implemented via the delay layers.

Reply:

For clarity, we reformulated this sentence (page 6, line 362).

The network architecture as shown in Fig. 1 is a feed-forward multi-layer configuration with four input neurons, followed by a variable-size neuron layer (hidden layer) and finally an output layer, comprising three neurons for the three classes (a study on deeper networks is provided in Section SI.A.1). Delay layers are inserted between neuron layers, as previously illustrated in the computational graph (Fig. 2).

Reviewer Point P.1.15 — ‘ensures that the delays remains bounded’ → ‘ensures that the delays remain bounded’

Reply:

Thanks, fixed.

Reviewer Point P.1.16 — ‘a mix a of both weights’ → ‘a mix of both weights’

Reply:

Thanks, fixed.

Reviewer Point P.1.17 — ‘parrot neurons’ nice use of underlying substrate

Reply:

Thanks!

Reviewer Point P.1.18 — 'sources of noise sources' → 'sources of noise'

Reply:

Thanks, fixed.

Reviewer Point P.1.19 — "However, as pointed out in [22], the finite-difference approximation appears to not be sufficiently accurate to achieve an improvement over fixed, random delays." This statement seems to only be considering delay-based surrogate gradients. In the citation above, synfire-gating information coordination is able to use surrogate gradients in a different manner to get MNIST accuracies of 97%.

Reply:

We have adapted the sentence (page 10, line 606) to make it clearer that it (and the reference [22]) specifically addresses gradients of *delays* calculated via the finite difference method employed in [19].

Reviewer Point P.1.20 — Additionally, the need for chip-in-the-loop training indicates that the actual training algorithm needs a standard computer to implement. Will it be possible to transfer exact gradients to a fully on-chip algorithm? This will be important since one of the most significant uses of neuromorphic AI is energy-efficient computation. The use of a standard computer for training effectively eliminates any efficiency in the training phase. And there are many off-chip learned neuromorphic SNNs that perform very well in the inference stage.

Reply:

This is related to our answer to Point P.1.1. Although it has not been demonstrated so far on hardware, there is nothing that fundamentally prevents the implementation of DelGrad or its weight-only predecessor [45] in a fully on-chip learning setting. In fact, the supplementary materials of [45] already contain a potential simplification of the gradient calculations that could lessen the computational demands for on-chip learning.

As an alternative to on-chip learning, the idea of first training a network model in software and only afterward porting it to a neuromorphic chip is often discussed in literature. However, typically this is only possible for digital chips without significant performance penalties, and even there the porting may be imperfect because of architecture-specific effects that were not considered during the simulation. For mixed-signal chips, the variability of the analog components renders such a straightforward transfer all but impossible. This is the main reason why chip-in-the-loop training is the standard procedure for such substrates.

Reviewer Point P.1.21 — Have the authors studied how deep of a network can be trained given their hardware noise? The BSS-2 chip may not be big enough, but this could be tested in simulation.

Reply:

We found this to be a very interesting point and have simulated deeper networks of two hidden layers with 30 neurons and axonal delays in ideal and hardware-aware simulations. Figure 1 shows that the additional layer in the network, despite the noise, can be utilized by

Figure 1: Comparison of test errors for networks with one and two hidden layers of 30 neurons each, both trained with axonal delays in ideal and hardware-aware simulations. The first two bars are ideal, the last two bars hardware-aware simulations. First and third bar are networks with one hidden layer, second and fourth bar are networks with two hidden layers. The numbers in parentheses indicate the number of trainable parameters in the network.

the algorithm to lower the test error. Note that the networks hyperparameters are the same as in Tables SI.2 and SI.3 with a slight modification of the loss $\Delta_t = 0.5$ for the hardware-aware simulation with two hidden layers.

Additionally, to address Point P.2.2 about the gradient flow in much deeper networks in simulation, we have added a new supplementary section (Section SI.A.1) and figure (Fig. SI.1). There, we show that, in an ideal simulation, a network with 5 small hidden layers achieves the same performance as a network with one big hidden layer and the same number of parameters. This demonstrates that our method is able to provide useful gradients for weights and delays in all layers and uses the available resources in the network fully.

Reviewer 2

The paper focuses on training transmission delays in networks of spiking neurons with exact gradient methods, using a framework of leaky integrate-and-fire neurons and Time-to-First-Spike coding (TTFS). This is a timely topic for both theory and hardware people and the paper contributes insights for both communities. The authors show that a situation where the membrane time constant is exactly twice the synaptic time constant is of particular interest because in this case the output spike time in layer n is an analytical function of the input spike times of neurons in layer $n-1$ (Eq. 5 in the paper). This explicit formula avoids access to the derivatives of the membrane potential at threshold (and also avoids the somewhat awkward Lambert function of Eq. 4).

Overall, I like the paper. In particular I am impressed by the training results on the Brainscales hardware platform with the hardware in the loop. The idea of introducing parrot neurons to implement delays on the existing neuromorphic hardware is nice. I suggest to publish the paper after my major comment and a few minor comments have been taken into account.

Reviewer Point P.2.1 — Major: The main text should stress more clearly and early on two assumptions made in the paper.

(i) I understand that the formalism can be extended to multiple spikes per neurons (as mentioned at the end of the Discussion). However, as the paper currently stands, all explanations and

notations in the theory part of the main text assume single-spike coding/Time-To-First-Spike (TTFS) coding, not only in the output layer but also in all intermediate layers. Otherwise Eqs (3-5) do not work. For example in equation (3), there should also be a dependence upon that last time of reset of the neuron in layer n ; and if the input neurons in layer $n-1$ are allowed to fire several spikes then the authors should mention more explicitly that the index i of t_i and w_i does not refer to neurons but to spikes so that a neuron with two spikes enters twice. (The assumption that refractory time is set to infinity is unfortunately only introduced in the simulation section. If the authors move this information to the theory section, the above problem will be solved).

Reply:

Indeed, we have chosen the single-spike formulation in the main text for simplicity, but that was not stated clearly enough in our previous version. We have now added a section in the supplementary material that details the equations for the multi-spike case (Section SI.D) and we reference this section in the main text (page 3, line 197; page 4, line 216), while keeping the single-spike case there for clarity.

Reviewer Point P.2.2 — (ii) The simulations are limited to networks with one hidden layer. This is a severe limitation. In networks with multiple hidden layers training instabilities may arise, which can be traced back to the vanishing or exploding gradient problem as suggested in Stanojevic et al. (2024). I assume that these instabilities can be controlled if gradients in the present model are evaluated AS IF the membrane potential was crossed with a slope of one. I guess, that, analogous to Eq. (5), the authors would be able to explicitly calculate the slope of the membrane potential at threshold and use this information to stabilize their algorithm for deep networks, analogous to Stanojevic et al. (The above problem could be simply solved by adding a few sentences in the discussion section on the generalization to multiple layers)

Reply:

The validation of our training method on deeper networks is indeed an important test. We have performed additional experiments with deeper networks and have added the results to a new section in the supplementary materials (Section SI.A.1) that is referenced where we first talk about network architecture (page 6, line 367). In summary, we show (Fig. SI.1) that a network with 5 hidden layers and a wider network with a single hidden layer with approximately the same amount of trainable parameters achieve the same performance. Our results demonstrate that useful gradients reach the lowest layer and we are able to make use of all trainable resources in the deeper layers.

In response to a question by another reviewer (see Point P.1.21) we have also looked into multi-layer results for the hardware-aware simulations, see Figure 1. In addition to that we have previously successfully trained weight-only networks (i.e., based on the DelGrad predecessor [45]) with 8 hidden layers on the MNIST dataset for a different project. This is not published yet, but the results can be found here: https://github.com/JulianGoeltz/fastAndDeep/blob/WeHaveToGoDeeper/experiment_pretrained/mnist_eightlayer_seed1/epoch_200/mnist_summary_plot.png

Reviewer Point P.2.3 — Minor comments (mostly linked to the above points) p1 less number → a smaller number

Reply:

Thanks, fixed with 'fewer'.

Reviewer Point P.2.4 — p2/Fig 1, Caption c: Please mention that ALL neurons in the hidden layers also fire at most a single spike, which is necessary for Eqs 3-5. Therefore not just the spike output layers is TTFS (in the sense that the first spike across all neurons counts) but each neuron in each layer has TTFS coding (in the sense that only the first spike of each neuron counts and later spikes of the same neuron are absent or ignored or suppressed by infinite refractory period).

Reply:

We agree that mentioning TTFS specifically for the decoding of the output neurons is misleading, thus we have clarified the used setup in the caption of Figure 1:

As sketched in the raster plot, in the experiments in this manuscript we employ TTFS coding, i.e., each neuron spikes only once, however our method also generalizes to multi-spike scenarios (Section SI.D) if required by the task.

Reviewer Point P.2.5 — p2 "resulting in selecting the suitable delays. " An early example of the idea of delay tuning via selection is 'A neuronal learning rule for sub-millisecond temporal coding.' Nature 383 (6595), pp. 76–78 (though in an unsupervised setting rather than a supervised one).

Reply:

Thank you for the nice reference, it has been added (page 2, line 85).

Reviewer Point P.2.6 — p3 "that employ exact spike time gradients [41, 42]. " May be the paper of Stanojevic which is cited later could already cited here? I leave this decision to the authors.

Reply:

We have added it to the citations there (page 3, line 171).

Reviewer Point P.2.7 — p3. before Eq. 3 For one such output spike time $T \rightarrow$ REPLACE BY: Assuming that the membrane potentials are all reset at time zero, the first output spike of neuron j in the next layer ...

Reply:

As mentioned above (Point P.2.1), we have made the single-spike scenario explicit, and discuss the more complex multi-spike case in detail, including the treatment of different boundary conditions for the membrane potential (page 24, line 1191), in the appendix (Section SI.D). Before Eq. (3) (page 3, line 197) we now explicitly make the distinction of the different cases.

Reviewer Point P.2.8 — p5. "The refractory period τ_{ref} is set to infinity, such that all neurons only spike once." THIS assumption is used much earlier (see above) so that the sentence should be shifted to the theory section.

Reply:

Our changes in response to Points P.2.1 and P.2.7 (see page 3, line 197) should ensure that the single spike simplification is clearly stated much earlier in the manuscript.

Reviewer 3

Summary:

The paper describes a method to train a feed-forward network of spiking neurons with delay. It defines a mathematical model for training on a general purpose computer, and an implementation on the mixed-signal brain scales neuromorphic hardware. The mathematical model computes analytically the exact spike-time for each neuron and adds a constant delay parameter. Backpropagation is therefore exact (as long as spiking order does not change, and no spike appears or disappears). For the hardware implementation, hardware in the loop training is used to train robustly despite imperfections of the semi-analog brain scales hardware. A simple noise model simulation is proposed to explain the reduction in accuracy after training on the hardware.

Main strength:

Developing a mathematical model that is viable for analog (or mixed hardware) is very difficult since it requires adapting to the hardware constraints. It is here well executed, and the achievement is well needed for the community since there are emerging techniques to train SNN using exact spike times but this is the first (or maybe second) demonstration on hardware which scales. Without a doubt it is the first to do that with delay parameters and prove on the hardware that delays can be used (and are useful) for a hardware implementation.

The writing is quite good, all the results are very clearly described, and I am convinced that the results are reproducible.

Reviewer Point P.3.1 — Main weaknesses: 1) Unfortunately the dataset is rather small and easy (probably significantly easier than MNIST for instance). It means that the analysis (synaptic versus axonal etc...) is probably not very informative on larger problems where depth would create more problems. Keeping the dataset small is totally understandable for the hardware implementation since the hardware dimensions are adding constraints, but I wonder if the claim that delay parameters are bringing benefits would be also true on a deeper network for instance (this is a very general speculation see comments 3 and 4, for more precise comments). To be clear I am not asking to try another dataset in a potential revision, rather, acknowledging that the dataset is small but adapted to the hardware prototype is sufficient. In this logic, any general claim on the quality of the mathematical model for neural network training should have a caution that it is only shown here for a small network and dataset.

Reply:

We are happy to see your appreciation of prototype hardware demonstrations and absolutely agree with your comments. Indeed, the YY dataset is rather small, and that is by design. The purpose of the dataset is to allow smaller-scale studies and hardware prototyping, as we do here – and, in particular, under the constraints imposed by our neuromorphic device, as you say. At the same time, the YY dataset is difficult in an important way. Unlike MNIST, where the interesting performance range (i.e., better than a linear classifier) starts at 92%, for YY it starts already at 70% [48]. We have elaborated on these points in page 6, line 350. In the outlook of the manuscript (page 10, line 683), we have highlighted the need to look at more complex datasets, especially ones with more temporal structure like datasets on speech recognition.

In this work, the YY dataset was used as a proof of concept and first step to benchmark our approach. The YY dataset provides a problem that can not be solved linearly and where the information can be presented using a TTFS encoding,

similar to the previous work [45]. As indicated by our experiments in Fig. 4d, the performance boost provided by the inclusion of learnable delays increases when the temporal features of the data span larger time scales.

Therefore, the natural next step will reside in a more thorough benchmarking on larger datasets, and in particular on data with explicit temporal components, such as [64, 65]. Especially for data provided by event-based sensors, longer time scales are required and TTFS might reach its limits as a feasible coding scheme.

The advantage provided by delays is corroborated for more general tasks in bigger networks by [18, Fig. 1c] and [22]. Additionally, the presence of delays in the top ranking performances on the leaderboard for the SHD dataset [64] also supports this conclusion.

2) I find that the intro is not doing a very good job at positioning this paper in comparison with the literature. Reading the intro it is sometimes ambiguous if the authors claim that exact backpropagation in spiking is a novelty of this paper for instance. Here are recommendations:

Reviewer Point P.3.2 — 2a) Conceptually, exact backprop for SNN was published in Bohte 2002. This ref is present but it should also be cited along with [41] and [42] in the intro.

Reply:

We agree and have added it there.

Reviewer Point P.3.3 — I find it interesting that in 20 years, this technique was forgotten and rediscovered. I wonder if the authors have something to add about that. Maybe it works better now because the neuron model changed? the dataset encoding? (Caution: I expect general speculation here, it is better not writing more rather than a half-convincing statement. For instance, it's not just about delays, given that many papers have shown that harder datasets are solvable without delay).

Reply:

We speculate that the forgetting of the techniques described by [24] is not due to specific details of the method compared to later ones, but rather due to unfortunate timing. That work was published before the deep learning revolution and far before the rising interest of the SNN- and neuromorphic community in deep learning methods. Additionally, we think that it suffered from a typically underestimated phenomenon: The reach/impact of a method or algorithm is not only determined by the actual qualities of said method but also to a significant portion by how well it fits to the commonly available and prevalently used computing hardware and frameworks that are available at the time (“hardware lottery”¹): [24] was published before wide-spread availability of CUDA support for GPUs, before the development of advanced deep learning computing frameworks like PyTorch and before the availability and adoption of common benchmarking frameworks. It was therefore significantly more difficult to reproduce or adopt by the community at the time than later discoveries that address similar questions.

Reviewer Point P.3.4 — 2b) The intro often refers to “the true gradient” but fails to report that there are many discontinuities in the network computation (example: after a weight update it probably occurs often that the output spike of a neuron arrives before it's input, or that a new spike is added). It would be great to acknowledge that the gradient computation,

¹<https://arxiv.org/pdf/2009.06489>

like others, is not solving this. Optionally: writing that surrogate gradient is not only inexact but also complementary since it addresses this partially, could be useful.

Reply:

You are correct that our approach does not directly treat the discontinuities due to the (dis)appearance of spikes: As you accurately pointed out, when we calculate a gradient and perform an update, there is a chance that the parameter change is strong enough to remove existing spikes or create new ones. Crucially, this is very much tied to the finite step size of the updates we take, and the fact that an infinitesimal, locally exact gradient does not predict accurate changes for finite update steps is also true in classical ANNs. Of course, this issue is more prevalent in SNNs, where the discontinuities stem from jumps in the loss itself (when a spike appears or vanishes), independent of the specific method of updating parameters. Surrogate gradients are an attempt to address this problem. It is a good suggestion to point this out in the manuscript along with a newly published reference which elucidates some properties and also shortcomings of surrogate gradients.

The underlying surrogate-gradient approach partially addresses, at the expense of both exactness and the conservative property the gradient field [29], the discontinuities of (dis)appearing spikes by smoothing out the spiking threshold.

Reviewer Point P.3.5 — 2c) The last paragraph of the intro: “In this work, we present DelGrad ...” would deserve some clarifications. When parsing the first sentence it is hard to know what the authors claim as the theoretical novelty. Is that the intersection of exact spiking and application to parametrized delay? or is it the demonstration that it can work on hardware (which I find a lot more impressive). This claim on memory foot-print is also not very convincing since the dataset size is meant to match the hardware size, so the CPU / GPU simulation is certainly not in a regime where memory foot-print is a real bottleneck, and therefore I would not be convinced that the authors have strong results to support that. In my opinion, it would be wiser to highlight that the study describes a dense analysis of different implementation strategies for delays in SNN and focus on the hardware part – rather than claiming strong general software results which are weakly supported.

Reply:

We agree that the hardware aspect is a central contribution of this work. To reflect this more clearly, we have updated the title to: “DelGrad: Exact event-based gradients for training delays and weights on spiking neuromorphic hardware.”

We have also clarified the last paragraph of the introduction (page 3, line 125) to better articulate that DelGrad was developed through an algorithm-hardware co-design approach, meaning that the algorithm was explicitly constructed with neuromorphic substrates in mind. This includes features like the real-valued nature of spike times on analog systems and constraints such as low I/O bandwidth and limited on-chip memory.

Therefore, we argue that the algorithmic formulation itself is one of the key results. By representing spikes as continuously-valued points in time rather than as binary vectors over discrete time steps, exact and event-based gradients for delays appear naturally. This change of representation enables a straightforward co-training of delays and weights, and allows a thorough analysis of the different types of delays. Crucially, this spike-time-based formulation also makes the hardware implementation more practical, as it avoids the need to communicate dense membrane voltage traces at every time step (see also the discussion, page 10, line 658), which is typically required by methods relying on surrogate gradients. Furthermore, co-

training delays and weights enables a reduction in the number of trainable parameters, and thus reduces the memory footprint on chip.

Regarding memory footprint specifically, we have clarified that our claim refers to model size, not the GPU/CPU memory of a simulation. On neuromorphic hardware, parameters are stored on-chip, and reducing their number directly lowers memory requirements. We also acknowledge the small size of the dataset we used, and this point is addressed in more detail in our reply to Point P.3.1.

Reviewer Point P.3.6 — Main questions or suggestions: 3) There seems to be an interesting emerging property of axonal delays. I suspect that adding a constant axonal to all neuron would have only benefits. It does not affect the gradient with respect to d or lower layers, but it avoids some of the gradient discontinuities: with sufficiently large axonal delay, the output neuron spike would never arrive before its inputs, thus avoiding at least this type of discontinuities. It seems that the authors did not try to add a global fixed axonal delay (not trainable), but I am guessing that the delays are currently initialized randomly (uniformly?) in the reachable interval $[0, \lambda * \tau]$. So with large λ , this effect could still emerge when most delays are large enough. This could explain that large λ stabilizes the training for all cases in Figure 4d (it seems true for all values of τ). Therefore, I wonder if that benefit would also be happening if axonal delays are simply lower bounded, or all having the same high constant value (not necessarily trainable)? If yes I wonder if the gap in performance brought by delays could be explained by that effect. Or is there another (and better supported) intuition on why delays are helping? Making new simulations with lower bounded delays (or constant) might be difficult for the authors, so discussing this verbally (at least in the rebuttal) might be convincing enough for me.

Reply:

The ODE of LIF neurons is independent of time, their dynamics are therefore time-shift-invariant. This means that a constant offset to the input times, identical for all neurons in a layer, will not change the dynamics but only shift the output spike times by the same amount. As a consequence, in our setting, a constant global axonal delay offset \tilde{d} would not have any functional effect: the ℓ th layer would receive its inputs delayed by the offset $(\ell - 1)\tilde{d}$, and spikes sets $\{t_i\}_\ell$ would just be pulled further apart from each other. The time-shift invariance of the loss function means that the gradients do not change either.

To elaborate more on the 'different nature of delay parameters', we think that a delay parameter allows bridging longer time gaps more easily and reliably: As shown in Fig. 4d, delays become increasingly helpful as the dataset span increases. For larger temporal differences, input spike times may be so far apart that the PSPs, whose width is determined by τ , may no longer overlap. In this case, classification without delays becomes more difficult. However, trainable delays can push these PSPs together again, allowing further fine-tuning of spike times through synaptic weights.

Reviewer Point P.3.7 — As a related side comment, I wonder if the influence of λ would be more readable in Figure 4d if the x-axis would be λ and the color code would be τ . Currently the shades of orange are all overlapping.

Reply:

We have tried it out and the resulting plot is shown below. It is important for us to keep the orange coloring, as we, throughout the whole paper, color any result that is produced

with axonal delays in orange. Comparing the two plots presenting the same data points, we believe that the original one is better suited for conveying the interplay of dataset span vs. delay range λ .

Reviewer Point P.3.8 — 4) I enjoyed the rationale that chip area savings makes axonal/dendritic delays better than synaptic delay. I also read that delay (refractory period) is quantized on 8-bit and weight strength quantized on 6-bit. Is it because neuron parameters can take more chip area and have therefore more precision? I also wonder, the voltage threshold or neuron bias is also quantized on 8-bit? I wonder whether there is something inherent in the physics of “hardware delay computation” that could make that encoding/implementation better than a weight encoding. I have no idea about this, but I would love to hear whether some different scaling rule emerge in terms of chip-area versus accuracy trade-off between delays and synaptic weights or threshold parameter.

Reply:

In general, there are multiple options to implement delays in a neuromorphic system, and the possible solutions span both purely digital and analog implementations. Axonal delay circuits are interesting from two perspectives: First, and as you correctly noted, an axonal solution may be allowed to occupy more space than a per-synapse one, as it impacts total system silicon area only linearly in the number of neurons (as opposed to the approximately quadratic scaling of a per-synapse implementation). This might then, indeed, also allow for an increased resolution of the delay parameter. Second, axonal delay implementations can potentially re-use some of the circuitry or logic (e.g., a digital counter) already in place for controlling the refractory period. In a future revision of BrainScaleS-2, true axonal delays could be implemented through a minor addition to the neuron control logic, then allowing for delay values up to the neuron’s refractory period, without a significant change in the area or energy footprint.

A delay parameter with linear control over the resulting shift of spike times can, indeed, profit from lower resolution requirements compared to, e.g., a weight parameter with highly non-linear control. This qualitative difference is already visible in Figure 4c and already this reduction of network parameters alone directly correlates with a reduced silicon footprint. Further savings beyond that are, of course, conceivable, but highly depend on the specific

implementation. Until such a future implementation is realized, we are bound to using the existing substrate with the design choices (e.g., parameter resolution) made in the past.

Reviewer Point P.3.9 — 5) Why stop around 35 hidden neurons if the crossbar array has a 256 x 256 size ? Any reasons to explain these dimensions would be useful.

Reply:

Due to the kernel trick, the Yin-Yang dataset becomes too easily solvable when larger hidden layers are employed (see Fig 2 and Table 1 in [48]). Therefore, if we want to demonstrate the propagation of useful gradients beyond the readout weights and into deeper layers, we need to use smaller networks where this propagation becomes necessary for good performance.

(Remarks on code availability) The code is readable and seems consistent with the reported results. I did not run the code.

Additional changes

In addition to the changes listed above, we have performed some select changes that we want to list here for transparency:

- The plotting style in Fig. 3c was adapted for better visibility of the different lines.
- We have noticed that the scales in Fig. 4b,c were inconsistent (linear and logarithmic), we have made them consistent.
- We added two names to the acknowledgments.
- In the appendix, we put Section SI.C closer to the other mathematical sections, and slightly adapted the content.
- In Figure SI.4d, we highlighted the correspondence between the spread of the sketched lines and the standard deviation, as well as the fact that fluctuations can lead to the disappearance of spikes.
- We rewrote Section SI.E to give more attention to details and be more easily comprehensible.
- For reproducibility we have added a supplementary section with all the hyperparameter tables.

Response to the reviewers

Reviewer 2

Overall, the authors addressed the points I had raised. Regarding the details, I still have three minor problems.

Reviewer Point P.2.1 — 1) For some reason, the citation to the paper of Stanojevic did NOT make it onto page 3 line 178 as promised by the authors in their response to referee.

Reply:

Thanks for noting this, we want to offer our apologies for not including the reference: we absolutely planned to do it but somehow the change must have gotten lost. We have now included it at this position.

Reviewer Point P.2.2 — 2) In caption of Fig. SI8 it should probably read horizontal and not vertical bar.

Reply:

Very good observation, we corrected this, thanks.

Reviewer Point P.2.3 — 3) The unpublished github reference in the reply to referee 2 DOES NOT show that the gradient travels back to the first layer. First, MNIST is known to be supereasy as a task - multiple layers are not needed. (see, e.g., Illing et al., 2019) Second, and more importantly here, if you overparameterize each hidden layer, you are in the so-called LAZY regime which is the regime theoretically explored in the paper on 'Neural Tangent Kernels' (NTK). Essentially, the theory says that in an overparameterized network the representation at initialization is rich enough that you only need to move a few weights by a small amount to learn the desired task. One way to do this, is to move only the output weights.

You can avoid lazy networks by choosing a very weak initialization (close to zero) so that many weights have to move. Lots of literature on that from the theory community. You can start with the paper of van Meegen and Sompolinsky (2025) or start from Saxe and Summerfield and look at papers cited there.

If you want to show learning in deep networks, you have to use tasks where depth is known to be important.

lazy and non-lazy:

Jacot A, Gabriel F, Hongler C. Neural tangent kernel: Convergence and generalization in neural networks. 2018. pp. 8571–8580.

Chizat L, Oyallon E, Bach F. On Lazy Training in Differentiable Programming. NeurIPS. 2018. Available: <http://arxiv.org/abs/1812.07956>

Flesch, ... Saxe, Summerfield, Neuron doi: 10.1016/j.neuron.2022.01.005

Meegen, Sompolinsky, <https://www.nature.com/articles/s41467-025-58276-6>

Illing et al. 2019 <https://www.sciencedirect.com/science/article/pii/S0893608019301741?via=ihub>

Reply:

Thanks for the references. Your summary of the provided literature sounds indeed interesting and we will study it for our future work, especially when investigating learning algorithms on MNIST.

Reviewer 3

I would like to thank the authors for addressing all my comments seriously. I was already favorable to the publication of this article and I continue to think it will be useful for the community.

Thanks for these nice words, they are appreciated.